# In-Situ Testing of a Multi-Band Software-Defined Radio Platform in a Mixed-Field Irradiation Environment

**Jan Budroweit** [1,*] , **Steffen Mueller** [2,†] , **Mattis Jaksch** [1,†] , **Rubén Garcia Alía** [3,†] , **Andrea Coronetti** [3,†] **and Alexander Koelpin** [4,†]

1   Avionic Systems, German Aeorspace Center (DLR), Institute for Space Systems, Robert-Hooke-Str. 7, 28359 Bremen, Germany; mattis.jaksch@dlr.de
2   Institute for Electronics Engineering (LTE), Friedrich-Alexander University Erlangen Nuernberg (FAU), Cauerstr. 9, 91058 Erlangen, Germany; ste.mueller@fau.de
3   European Organization for Nuclear Research, CH-1211 Geneve 23, Switzerland; ruben.garcia.alia@cern.ch (R.G.A.); andrea.coronetti@cern.ch (A.C.)
4   Electronics and Sensor Systems, Brandenburg University of Technology Cottbus-Senftenberg (BTU), Siemens-Halske-Ring 14, 03046 Cottbus, Germany; alexander.koelpin@b-tu.de
*   Correspondence: jan.budroweit@dlr.de; Tel.: +49-421-24420-1297
†   These authors contributed equally to this work.

**Abstract:** This paper presents an in-situ test concept for a multi-band software-defined radio (SDR) platform in a mixed-field radiation environment. Special focus is given to the complex automated test setup with respect to the requirements of the irradiation facility. Additionally, selected test results of a system-level evaluation are presented and discussed. For the verification of the mixed-field radiation environment, the software-defined radio (SDR) was also tested under proton irradiation. The cross-sections for the observed single event effects are compared and show similar results.

**Keywords:** in-situ testing; automated test setup; mixed-field irradiation; total dose effects; single event effects; software-defined radio

## 1. Introduction

A software-defined radio (SDR) is a system that allows simple reconfiguration of the radio signal and radio frequency (RF) circuitry by changing the signal processing algorithm in a digital signal processor (DSP) or field programmable gate array (FPGA). The German Aerospace Center (DLR) has developed a new, highly-integrated generic SDR (GSDR) to realize multi-band operation on a single radio platform for spacecraft applications, using a state-of-the-art signal processing device and two radio frequency integrated circuit (RFIC) chips as programmable RF front-end [1]. The GSDR system is designed with a low-cost approach, avoiding implementation of expensive radiation hardened (RadHard) devices, but focusing on mitigation of radiation effects on both system and circuit levels. The system is tested in a unique radiation environment providing mixed-field radiation. The most challenging part of the irradiation test is the test setup of a highly automated test process with a limited number of available interfaces and radio frequency cable connections. In Section 2, the background and motivation is presented. The test purpose and requirements are presented in Section 3. The test bed concept and implementation is following in Section 4. Section 5 presents selected test results and the prediction of different in-orbit rates. Proton induced radiation effects test results on the GSDR are presented in Section 6 and are compared to the test results of the mixed-field radiation test.

## 2. Motivation and Background

### 2.1. Multi-Band Software-Defined Radio Platform for Space Applications

The GSDR system consists of a Zynq-7020 baseband processor and two AD9361 agile RF transceivers [2]. Two double data rate synchronous dynamic random access memory (DDR3-SDRAMs) are used to provide dynamic memories for the operating system. A NAND flash device stores boot images and sensitive data. For the Zynq-7020, DDR3 and NAND flash device being used in the system design, radiation tests have been performed by different institutions [3,4]. An on-board power distribution unit breaks the unregulated main input voltage down to 12 different sub-voltages.

A model of the GSDR hardware with its adaptable mother and daughterboard design is presented in Figure 1. The design of the GSDR is mainly based on commercial-off-the-shelf (COTS) devices, which mostly have already been investigated (and individually tested) for radiation effects before [5]. Critical system parts, like the input power regulator or digital interface driver of the GSDR are designed to be replaced with RadHard solutions. To improve the system reliability, sensitive devices, such as power regulator or SRAM-based memory, have been evaluated to their expected behavior and multiple mitigation techniques, e.g., device level single event latch-up (SEL) protection or memory scrubbing, have been implemented into the system design.

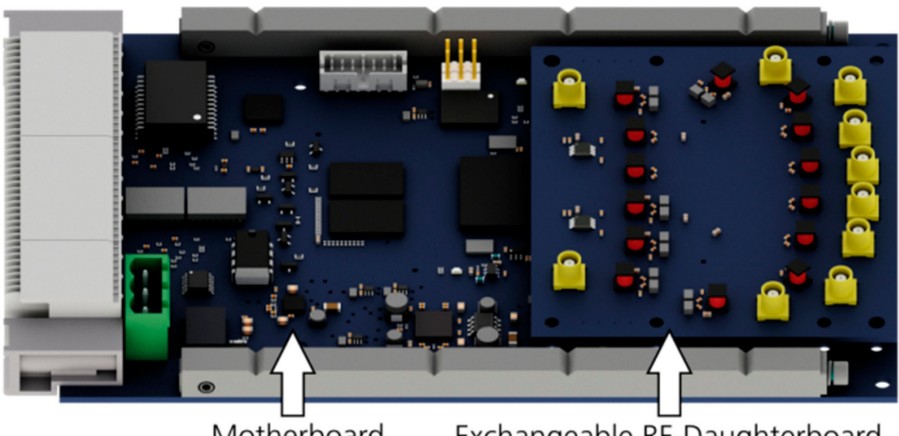

**Figure 1.** 3D-Model of the generic software-defined radio (GSDR) system.

### 2.2. CHARM—CERN High Energy Accelerator Mixed-Field Facility

CHARM is a radiation facility of the European Organization for Nuclear Research (CERN), which uses the 24 GeV proton beam provided by the proton synchrotron (PS) complex in order to generate a mixed-field radiation environment. The primary beam is extracted from the PS and is directed to different selectable metal targets (e.g., copper) to emulate multiple radiation spectra. A detailed description of the facility is given in [6]. Inside of CHARM, different positions can be selected to place the system under test (SUT), where each position provides a different radiation spectrum (refer to Figure 2).

The GSDR was located at positon PC0 (green box in Figure 2). The expected radiation spectrum is presented in Figure 3, as simulated using the FLUktuierende KAskade (FLUKA) Monte Carlo simulation tool. The test and measurement equipment is placed in the control room during the irradiation phase. The control room and the irradiation room are connected via patch panels providing several digital, analog, and RF cable connections.

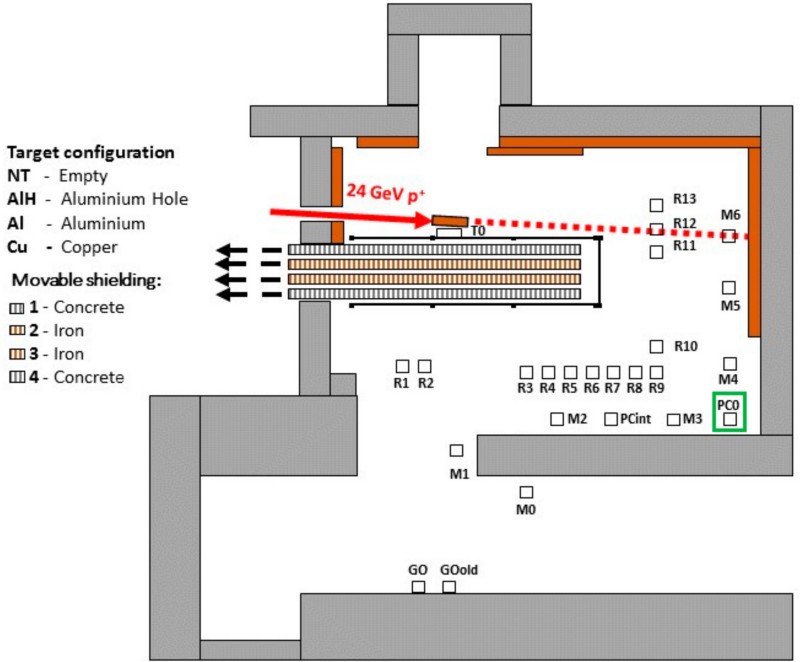

**Figure 2.** Layout of the irradiation area at CERN High Energy Accelerator Mixed-Field Facility (CHARM) test facility [6].

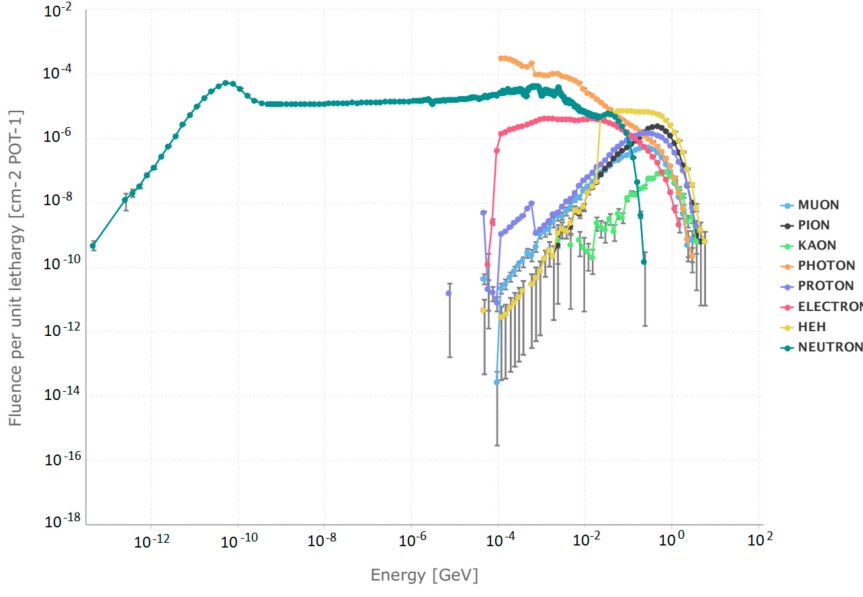

**Figure 3.** Spectra at position PC0 with copper target and no additional shielding.

As can be seen in Figure 3, multiple particle (#) species are present at CHARM over a broad energy range. However, those relevant for single event effects (SEE) induction are the hadrons, which are capable of generating nuclear interactions leading to localized energy deposition causing SEEs. In first approximation, all hadrons above 20 MeV (mainly neutrons, protons and pions) are considered as equality efficient in inducing SEEs, whereas below this energy, charged hadrons are disregarded due to their loss of energy in packaging materials and Coulomb repulsion with the nuclei near the sensitive SEE region, and neutrons are weighted with a response function in the 0.2–20 MeV range according to experimental single event upset (SEU) data for a given reference SRAM memory.

The sum of the hadron fluence above 20 MeV plus the intermediate energy (0.2–20 MeV) neutron contribution is defined as the equivalent high-energy hadron fluence and used to calculate soft-error cross sections in the mixed-field environment.

In terms of time structure, protons arrive to the CHARM target in spills lasting roughly 350ms and with a repetition period of roughly 10 s. Despite the pulsed nature, the spill duration is orders of magnitude larger that the SEE characteristic time scale, which is typically in the ns range, therefore irradiation can be considered as quasi-continuous. Nominal spills contain $4.5 \times 10^{11}$ protons, and for the PC0 location correspond to a high energy hadron (HEH) equivalent flux of ~$1.02 \times 10^7$ HEH/cm$^2$/spill and a TID of 0.44 rad(Si)/spill.

CHARM irradiation campaigns typically last for ~5 days. Provided accesses to the irradiation area are performed on a weekly basis.

## 3. Test Purpose and Test Bed Requirements

### 3.1. Test Purpose of the GSDR

The major aim of this test is the verification of the implemented mitigation techniques and evaluation of the overall performance on system level. Thus, different single event effects (SEE) and total ionizing dose (TID) effects are monitored and counted to analyze the robustness of the system in a radiation environment.

### 3.2. Requirements and Design Constraints

Since the purpose is the validation of the system functionality, it requires monitoring of power, data and RF interfaces. In order to perform two GSDRs with full functional RF operations, it requires monitoring of eight separated RF interfaces. Due to the limited numbers of low-loss RF cables at CHARM, another solution is required to meet these requirements. In addition, it needs to take into account that especially RF cabling requires a dedicated effort, since they are sensitive to temperature drifts, mechanical stress and imperfections which could lead to mismatches and degradation in RF signal performance. For this reason, an RF multiplexer (MUX) has been evaluated as the best solution with respect to time effort, complexity, and costs [7]. Due to the selective switching property of the MUX, full parallel testing of all RF interfaces of the GSDR is not possible, but a multi-band test approach is still feasible and adequate for system level testing. Using combiners would allow RF interface independent measurements, but would require band selection of each RF path. A detailed description of the test bed configuration and the RF MUX is presented in the following section IV.

## 4. Test Bed Concept and Implementation

### 4.1. Test Bed Schematic

Figure 4 presents the schematic of the GSDR test setup. On the left-hand side (white), the control room area is shown where the test equipment is placed. The right-hand side of Figure 4 (gray) represents the irradiation room with two SUTs and the RF MUX. Each SUT consists of two transmit-and-receive chains and uses a corresponding reference transceiver (REF) at the control room side for RF data transmission and reception. Both areas are connected via patch panels on each site and a series of cables in between [8].

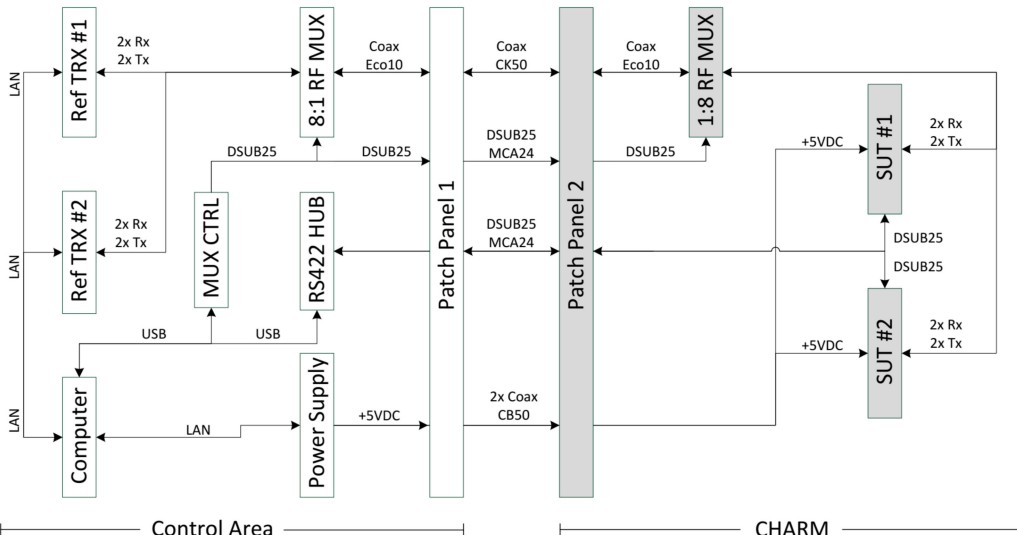

**Figure 4.** Schematic of the GSDR test setup.

All eight configurations can be accessed via appropriate MUX settings (e.g., SUT#1 transmission (TX) #1 to REF#1 reception (RX) #1). The full test equipment consists of:

- 2× reference transceiver systems (REF)
- 1× computer for test automation and signal processing
- 1× power supply unit to supply the SUTs
- 2× 1:8 MUX and 1× MUX control unit
- 1× RS422 HUB supporting the digital interface between the SUTs and the computer

Each device of the test equipment setup is connected via ethernet or USB to the main control computer. A more detailed description about the automatic test procedures and failure handling mechanism is presented later in this section.

## 4.2. RF Multiplexing and RF Path Characterization

The MUX is a key component of the test setup in order to enable efficient in-situ testing of the SUTs with respect to RF signal acquisition. Applying the concept of multiplexing to the GSDR test setup allows selective routing of 4 RX and 4 TX signals to the control room using a single coax cable connection, only. This significantly reduces both cabling effort and signal integrity issues compared to the classical 1-by-1 instrumentation approach that would require eight coax cables, instead.

As one MUX resides inside the irradiation room, it has to be RadHard in order to not corrupt the transmitted signals. Since there is no commercial RadHard MUX available, a custom design is carried out. Radiation hardness is achieved by both, design and technology [7].

The RF switches are based on electro-mechanical relay technology which is inherently RadHard. On the other hand, the entire control logic is separated in the MUX control unit that resides outside the irradiation locale. In order to yield good performance, an RF substrate is used that is based on hydrocarbon ceramic laminate (Rogers RO4350B). The latter is suited for low-loss coplanar waveguide structures with superior RF characteristics, while being free of polytetrafluoroethylene (PTFE) which could degrade the substrate's dielectric performance when being exposed to high TID levels. The MUX features a frequency range from DC to 6 GHz (Figure 5).

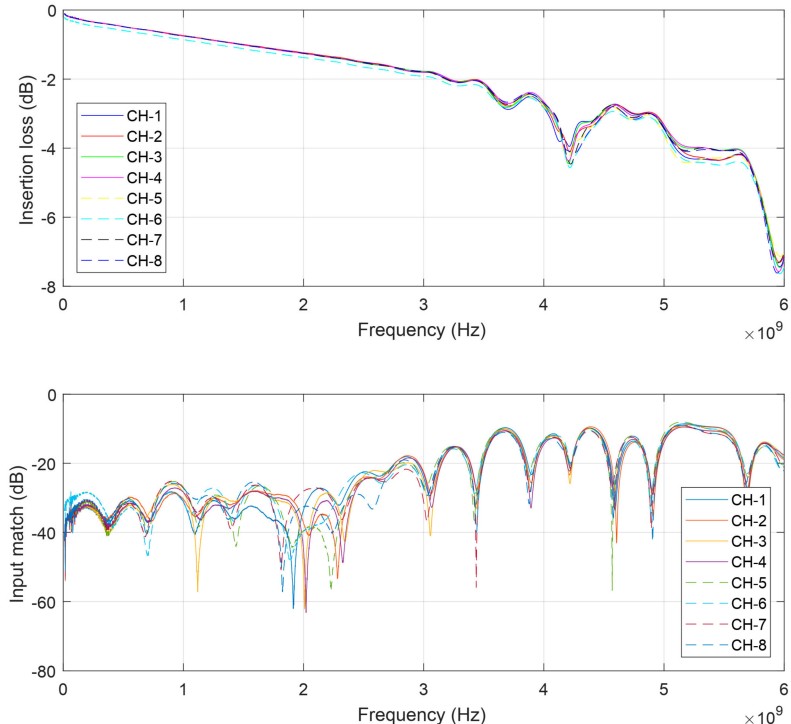

**Figure 5.** Insertion loss (**top plot**) and input match (**bottom plot**) of the MUX.

The transfer characteristics are especially linear up to 4 GHz. The insertion loss (S21) is −1.6 dB at 2.4 GHz and the input match (S11) is less than −25 dB. Thanks to a fully symmetric layout, a very high channel match could be achieved, i.e., wanted signals see the same transfer characteristics regardless of the MUX channel, which promotes reproducibility between single measurements.

*4.3. Implementation and Test Procedure*

In addition to the test setup schematic presented in Figure 4, there is a significant amount of software logic required to realize an automatic test process.

In Figure 6, the basic control and data flow is presented. The major unit that controls and processes the data is the on-board computer (OBC) located in the control room area. The OBC commands a repeating procedure for each SUT and receives the data via an UART/RS422 interface, and interacts with the power supply unit (PSU) and the REF. The OBC is also commanding the RF MUX to connect both SUTs to their corresponding REF. The repeating command procedure contains a request of housekeeping data and the execution of individual RF activities (capturing of received baseband data of the SUTs and the REF and further live-processing to waterfall fast-Fourier transformation (FTT) plots). For all RF activities, a known set of data will be transmitted at three different carrier frequencies (900 MHz, 2.4 GHz, and 5 GHz). The OBC follows the command procedures and is waiting for the SUTs responses. For the housekeeping data request, both SUTs are operating independently. Since both SUTs are sharing the RF MUX unit for their RF activities, the OBC is supervising the RF MUX allocation. If the currently allocated SUT has responded the data to the OBC, it will release the RF MUX and the next waiting SUT is able to request the MUX. In case an SUT is not responding after the third request on the same command, the OBC assumes that the SUT is malfunctioning and performs a power-cycle by triggering the SUT related PSU output.

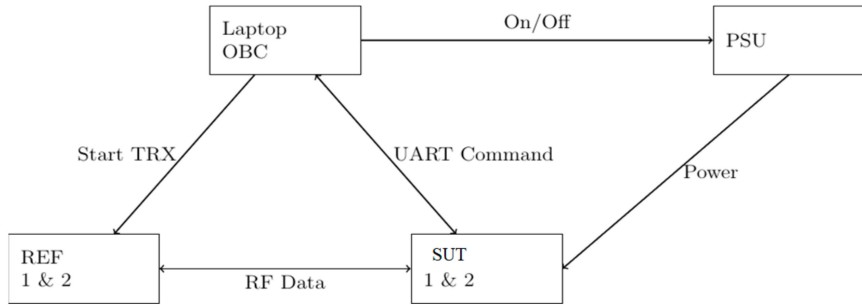

**Figure 6.** Control and data flow of the GSDR test setup.

The SUT is usually capable to handle a series of malfunctions by itself. An internal voltage and current monitoring is detecting SELs on each sub-voltage power line and will release an internal reboot if an SEL occurs. In case that the software crashes, a system hardware watchdog will also trigger a reboot. The watchdog requires a heartbeat signal from the processor (software) and if the signal disappears longer than 1.6 s, it enables a fault output flag and forces the system shutdown and reboot. The reboot process takes about 10 s until the SUT is operable and can execute and response to the commands of the OBC. Additionally, the SUT software is monitoring different system parameters (e.g., RFIC register configuration, boot medium configuration, or multiple software threads) and is able to handle soft errors without forcing a reboot.

## 5. Experimental Results

During the first spills malfunctions and resulting reboots on both SUTs were observed, that were triggered by the internal watchdog unit. In most cases these malfunctions were observed on the ARM processor of the Zynq and forced a kernel panic. In some cases, we also lost connection to the SUT without a trigger of the internal watchdog, leading into an external power-cycle by the OBC/PSU. In any case, interrupted boot-processes haven't noticed nor corrupted boot images, stored in the NAND flash, have been observed that requires a recovery. A summary of the average numbers of reboots and power-cycles for both SUT is given in Table 1. The total time of irradiation was approx. 112 h. During this time the SUT received a total high-energy hadron equivalent (HEHeq) fluence of $2.170 \times 10^{11}$ #/cm$^2$ and a TID of 9.25 krad(Si), as measured by local on-line monitoring by means of the RadMON system [9].

**Table 1.** Average reboot and power-cycle events and their cross-sections.

| Type | #Event | #Spills | HEHeq Fluence [/cm$^2$] | Cross-Section [Device/cm$^2$] |
|---|---|---|---|---|
| Reboot | 5320 | 21236 | $2.170 \times 10^{11}$ | $2.451 \times 10^{-8}$ |
| Power-cycle | 75 | 21236 | $2.170 \times 10^{11}$ | $3.456 \times 10^{-10}$ |

In addition to the general reboot and power-cycle events, the internal monitored voltages and currents of the housekeeping data set have been analyzed as presented in Figure 7.

Abnormalities in the voltage domain, neither any SEL or high current state has been observed. The ripple effects on the current domain are explained by the different RF activities on the RFICs and the required processing power of the Zynq. The post processing of the waterfall FFT plots, presented in Figure 8, also shows a similar pattern over full span of irradiation. Thus, a malfunction or degradation of the RFIC can be excluded. It has to be mentioned that the function of SUT#2 has been partially lost, due to a broken SD-Card (which was not readable anymore) at $1.570 \times 1011$ HEHeq/cm$^2$.

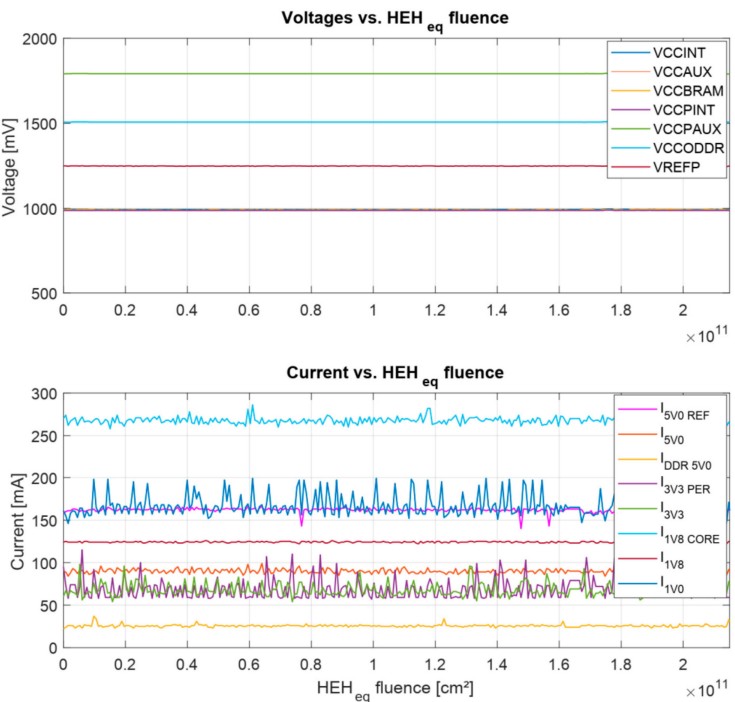

**Figure 7.** Monitored internal voltages (**top**) and currents (**bottom**) during irradiation of SUT#1.

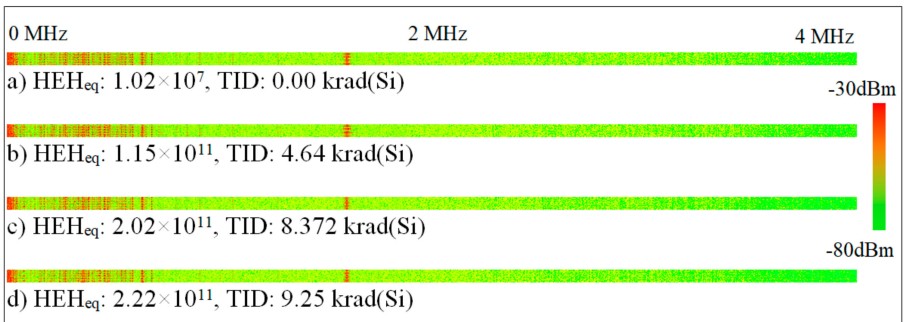

**Figure 8.** Waterfall fast frequency transformation (FFT) plot at 2.4 GHz center frequency on reception (RX) #1 of system under test (SUT) #1 for fluences (HEHeq) and TID values.

The SD-Card was partly used to store data, before the transmission to the OBC starts. Fortunately, for SUT#1 the SD-Card was functional all the time. Nevertheless, SUT#2 was always able to recover to operational mode, even if most of the data to be generated were not storable anymore. Up to the SD-Card malfunction of SUT#2, we observed similar event rates compared to SUT#1. Thus, we assume that the variations of the cross-sections between both SUTs are small enough.

In Table 2 the predicted in-orbit rates for sun-synchronous low earth orbit (LEO) and the international space station (ISS) are presented. The calculation and prediction model is based on the OMERE software (in version 5.2.4) [10].

Due to the mixed-field approach it is not possible to determine a straight energy threshold. Thus, the energy threshold as per definition of CHARM that all hadrons above 20 MeV are considered as equality efficient in inducing SEEs has been chosen. The cross-section saturation for both types of single event failure interrupt (SEFI)s are taken from Table 1. Heavy-ion rates, predicted by OMERE, are not considered. However, for the LEO and ISS orbits, protons are expected to dominate the overall soft error rate.

**Table 2.** Predicted in-orbit rates based on the test results (launch date: 6 June 2019).

| Orbit | SEFI Type | Proton Rate [failure/device/day] |
|---|---|---|
| LEO, 800 km, 98° | Reboot | $1.60 \times 10^{-3}$ |
| LEO, ISS | Reboot | $1.99 \times 10^{-4}$ |
| LEO, 800 km, 98° | Power-cycle | $2.26 \times 10^{-4}$ |
| LEO, ISS | Power-cycle | $2.80 \times 10^{-4}$ |

## 6. Comparison to Proton-Induced SEE Test Results

In order to verify the results from the CHARM test campaign, one of the SUTs has been irradiated with mono-energetic protons only at the Kernfysisch Versneller Instituut (KVI) located on the Zernike Campus of the University of Groningen, Netherlands. For this test purpose, the GSDR motherboard has been exposed to a proton beam with a primary energy of 184 MeV. Using a scalable degrader setup, the energy has been reduced down 150 MeV, 120 MeV, 100 MeV and 70 MeV. In this test configuration, lower energies were not feasible to test, due to large degrading of the primary beam energy and the resulting worse inhomogeneity. Moreover, degrading such high energy would also produce a lot of unwanted particles, such as neutrons, which might affect the expected test results.

Since the most sensitive parts of the GSDR observed at CHARM test are the Zynq and the dynamic memory storing the operation system (OS), the proton beam has been focused to the Zynq-7020, the DDR3-SDRAM and the NAND flash with a 50 mm circular collimator as illustrated in Figure 9. In this test configuration the exchangeable RF daughterboard has been demounted to enable full access to the components of interest for the proton beam. Further investigations on the RF data in this test campaign have been skipped due to the radiation test results that have been obtained previously on a separated proton test, focusing only on the AD9361 [11]. The GSDR was running the same software (except the usage of a SD-card for intermediate data storage and collecting RF data) and the numbers of events were counted. Due to the limited access time to the test facility, the target fluence has been reduced to $5.0 \times 10^{8}$ #/cm². The average flux depends on the selected proton energy and varies between $1.0 \times 10^{6}$ and $5.0 \times 10^{6}$ #/cm²/s in order to get a moderate event rate and being able to interact quickly with the beam control of the facility.

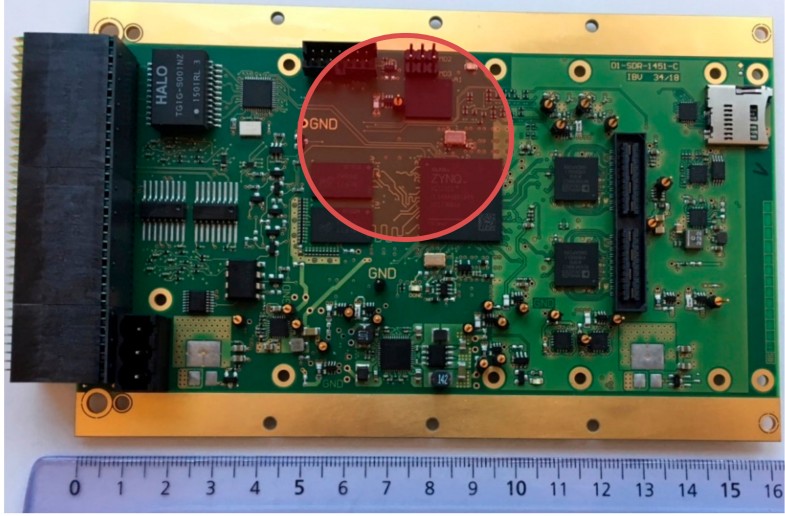

**Figure 9.** GSDR motherboard with highlighted radiation exposed area using a 50-mm circular collimator (covering Zynq-7020, DDR3-SDRAM and NAND flash).

The SEFIs are also separated into two categories: (1) Reboot: self-recovered SEFI by the GSDR internal watchdog and (2) Power-cycle (PC): external power-cycle to recover from a functional failure

of the GSDR which was not detected by the watchdog. The cross-section and counted events for the reboot SEFI events (1) are presented in Table 3.

**Table 3.** Reboot events and their cross-sections for proton-induced single event effect (SEE) characterization on the GSDR.

| Proton Energy [MeV] | #Event | Avg. Flux [#/cm$^2$/s] | Fluence [#/cm$^2$] | Cross-Section [cm$^2$/Device] |
|---|---|---|---|---|
| 70 | 7 | $1.1 \times 10^6$ | $5.0 \times 10^8$ | $1.4 \times 10^{-8}$ |
| 100 | 8 | $1.3 \times 10^6$ | $5.0 \times 10^8$ | $1.6 \times 10^{-8}$ |
| 120 | 8 | $5.1 \times 10^6$ | $5.0 \times 10^8$ | $1.6 \times 10^{-8}$ |
| 150 | 11 | $6.5 \times 10^6$ | $5.0 \times 10^8$ | $2.2 \times 10^{-8}$ |
| 184 | 13 | $5.0 \times 10^6$ | $5.0 \times 10^8$ | $2.6 \times 10^{-8}$ |

An illustration with the Weibull fitting curve of the SEFI reboot vs. energy is given in Figure 10. The Weibull fitting parameters W and S are estimated using the OMERE software (in version 5.2.4) [10]. The energy threshold is lowest being tested (70 MeV) and the saturation cross-section is ~$2.6 \times 10^{-8}$ cm$^2$/device. Power-cycle events were observed rarely (in order of maximum one event per run) due to the low target proton fluence. Thus, the results are not presented in detailed here and it becomes hard to make an accurate correlation with the cross-section for the CHARM results ($3.456 \times 10^{-10}$ cm$^2$/device) without further irradiation. However, the trend analysis is going to a similar cross-section value. As no interrupted reboot process has been observed at CHARM, the boot process of the SUT has been evaluated separately from the operational test scenario.

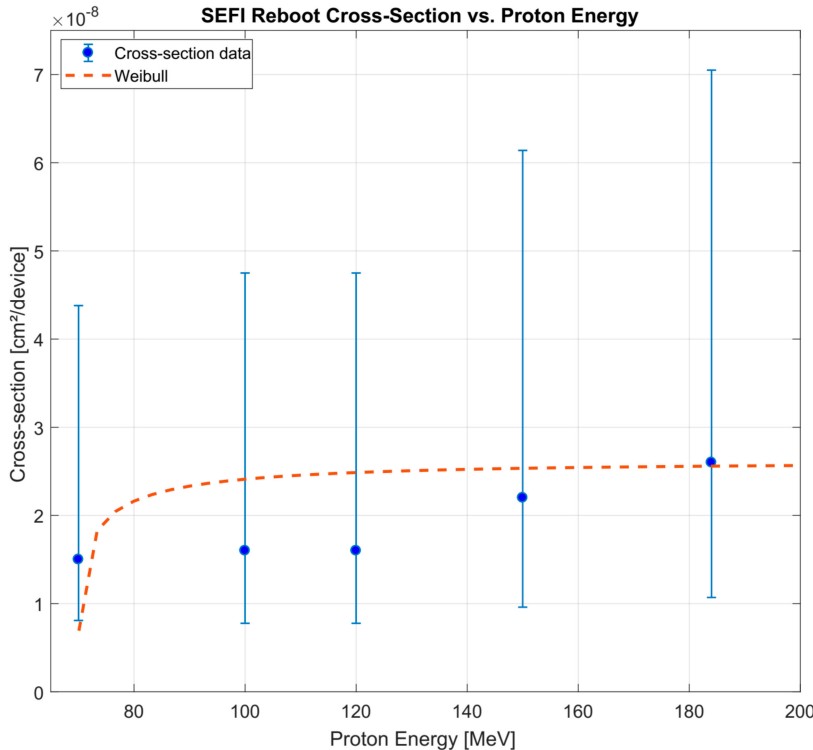

**Figure 10.** Cross-section for the SEFI reboot events vs. proton energy ($E_{th}$ = 70 Mev, W = 1.24, S = 0.439). Error bars are Poisson errors at 90% confidence level.

At least ten boot processes have been irradiated for proton energies for 70 MeV to 184 MeV. Neither boot interruptions, dead locks, nor other malfunctions have been observed.

In any case of the minor observed SEFI power-cycles, it has been observed that the system was partially functional, but was no longer responding to the request of the OBC anymore. Due to the

selected communication timeout, the OBC interprets a failure of the SUT and is triggering an external power-cycle by switching the PSU output off and on.

## 7. Conclusions

In this paper, the in-situ testing for a multi-band SDR platform in a mixed-field radiation environment has been presented. Due to the complexity of the SUT and the limited interfaces of CHARM facility, a novel test setup has been designed using a radiation-tolerant RF MUX. The experimental results were briefly discussed and show that no destructive SEE was observed for both SUTs. SEFIs were leading to high number of reboot events, but SUT was able to recovery any time without any interrupted boot processes or corrupted boot images. The SUT showed resistance to TID up to levels beyond typical specification for low earth orbit missions. Merely, the SD-Card of SUT#2 broke during the radiation test, which leads into limited measurement data. In future designs of the SDR, the use of SD-Cards is going to be avoided. However, the proof of concept for testing SDRs in radiation environment was successfully demonstrated. The SEFI rates have been used to predict in-orbit rates for different reference mission (GEO, LEO, ISS). In order to verify the CHARM test results, the GSDR has been tested under proton irradiation at KVI. The results show a similar cross-sections compared to the categorized SEFIs for the mixed-field radiation test at CHARM, serving as a confirmation of the representativeness of the latter in terms of performing soft-error rate in-orbit estimations for trapped protons environments.

It is worth highlighting that the CHARM facility enables the possibility of system-level testing for space, achieving weekly radiation levels for the configuration and location considered in this paper of $\sim 1 \times 10^{11}$ HEHeq/cm$^2$ and ~10.0 krad (Si), sufficient for most LEO orbit qualification requirements. Of course, challenges and limitations associated to CHARM qualification must be considered, such as the reduced irradiation accessibility (weekly access), the fact that any test equipment to be placed in the vicinity (i.e., several meters) of the system-under-test will also be subject to large radiation levels without the option of local shielding, or the lack of higher LET, as well as heavy ions in the environment.

**Author Contributions:** Design and concept, J.B.; test methodology, J.B. M.J., S.M., A.C., R.G.A.; software, M.J.; test preparation, J.B. M.J., S.M., A.C., R.G.A.; analysis, A.C., R.G.A.; investigation, X.X.; resources, A.C., R.G.A., A.K.; data curation, J.B. and M.J.; writing—original draft preparation, J.B.; A.C.; R.G.A.; writing—review and editing, All; supervision, R.G.A., A.K.

**Funding:** This work has received funding from the European Union's Horizon 2020 research and innovation program under the Marie-Sklodowska-Curie grant agreement number 721624—RADSAGA.

**Acknowledgments:** Special thanks go also to the CHARM team, Salvatore Danzeca, Chiara Cangialose and Jerome Lendaro for their support.

**Conflicts of Interest:** The authors declare no conflict of interest.

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
