# Peer review of "In-Situ Testing of a Multi-Band Software-Defined Radio Platform in a Mixed-Field Irradiation Environment"

_aerospace, doi:10.3390/aerospace6100106_

Round 1

Reviewer 1 Report

1. A few inconsistencies that should be easy to correct:
a. Be consistent in the use of Arabic vs Roman numerals. See line 44.
b. Be consistent in the use of periods vs commas to denote a decimal point.
c. Use the same radiation units. You have mixed rads and grays. Remember, the energy absorbed depends on the target, so you should specify rad(Si) or rad(SiO2) as appropriate.

2. Fix the caption for Table 1.

3. The waterfall plot in Figure 9 is not useful. A better way to plot that would be as a spectrum. Not only would the data be much easier to see, but the plot would show up better in B&W.

4. You have made no mention of dead time. How long does it take to reset after an event or to perform a power cycle? Consider the data in Table 3. For the 184 MeV protons, you have on average an event every 10 seconds. If the watchdog takes 1.6 seconds to detect an event, then you have at least 16 seconds out of 100 where you are not collecting data. This doesn’t include the actual time it takes to perform the reboot or power cycle. In other words, your cross sections is going to be higher than what you reported.

5. Let’s apply the same process now to the CHARM results. You have plenty of time between spills for resets and power cycles. However, the flux is quite high, so there is a possibility that multiple particles could cause a SEFI, but you would only detect one of those.  As you account for these effects, be sure to include a explanation in your paper.

6. In Figure 11, you need to include error bars. These are especially important when you have so few data points. Your Weibull may look different when you include error bars in your fit. Also, you need to justify your assumption that the threshold is the lowest energy you tested. In general, this is a poor assumption. Especially with protons where you have mixed effects from nuclear recoils and also direct ionization events.  Finally, it is not obvious why the data points in Fig 11 are different from those in Table 3.

7. This paper addresses only the basic functionality of the device and if it reboots. Perhaps of equal or even more importance to a satellite designer would be the errors introduced into the RF stream. For instance, do you get RF dropouts? If you send a data stream across the RF channels, what is the bit error rate?

Author Response

Dear Reviewer,

we would to thank you for your time and effort in reviewing our manuscript. Please find attached our reply to your comments. We hope that we covered all your comments to your satisfaction and revised the manuscript accordingly.

Thanks again

Reviewer 2 Report

Consider adding more background, prior art references to the introduction to improve context. Legends in Figs. 2, 3, and 8 are hard to read.  Consider improving resolution and/or font size to improve legibility. Page 7, Line 207: header for Table 1 is incorrect.  Consider revising. Page 9, Line 238: Neglecting heavy ion contributions for LEO is probably ok -- certainly for ISS, maybe for sun-sync depending on reliability & availability requirements.  Neglecting heavy ion contributions for GEO is probably inappropriate.  High-energy protons at GEO are limited to solar protons and the smaller content within the galactic cosmic ray spectrum.  It makes the results shown in Table 2 misleading for GEO operations.  Also, if more rad-hard components are added to the system, heavy ion contributions will become more important. Page 10, Line 268: A Weibull fit for these is less than optimal.  Traditionally, proton data are fit with a 1- or 2-parameter Bendel function.  A Weibull or Lognormal could be used, but more care has to be taken with the goodness of fit.

Author Response

(The authors gave the same response as above.)

Round 2

Reviewer 1 Report

My major concerns have been addressed.

I do have one other suggestion: In your abstract you state "Although the paper explains the details of this complexity, more can be discussed about the level of success achieved of the automated setup."  However, I think your paper should focus on the results, not the automation.  After all, automation in radiation experiments isn't something new, it's pretty much a requirement these days.  A description of your experimental setup is useful, but the audience of this journal will probably be more interested in the actual results.

Author Response

Dear Reviewer,

thanks for the review and your suggestion. We removed the sentence in the abstract to lead the focus a bit more to the results.

You are right that an automation should be state-of-the-art in radiation testing, just for the simple reason to safe time and thus money!

System-level testing is something that becomes more and more popular and where publications are currently quite rare. Depending on the system complexity, the radiation test setup will also be complex, since you want to to have an automated procedure for failure handling. The reason why we focused first on the setup was because that the initial paper was just focusing on the setup and the results were not analyzed in that detail and we also hadn't had the results from the proton test.

The paper content changed in time and the focus shifted a bit more to the results, which are indeed more interesting for the community. But since we spent a lot of effort and time in the test preparation we decided to keep also an eye on the complexity of a automated test setup for this system level test.

Thanks again

Jan Budroweit